# A Systematic Review of Medical Image Quality Assessment

**DOI:** 10.3390/jimaging11040100

**Published:** 2025-03-27

**Authors:** H. M. S. S. Herath, H. M. K. K. M. B. Herath, Nuwan Madusanka, Byeong-Il Lee

**Affiliations:** 1Department of Industry 4.0 Convergence Bionics Engineering, Pukyong National University, Busan 48513, Republic of Korea; sewmi96@pukyong.ac.kr (H.M.S.S.H.);; 2Digital Healthcare Research Center, Pukyong National University, Busan 48513, Republic of Korea; nuwanv@pknu.ac.kr; 3Division of Smart Healthcare, College of Information Technology and Convergence, Pukyong National University, Busan 48513, Republic of Korea

**Keywords:** medical image quality assessment (MIQA), imaging modalities, subjective assessment, objective assessment, artificial intelligence (AI), machine learning (ML)

## Abstract

Medical image quality assessment (MIQA) is vital in medical imaging and directly affects diagnosis, patient treatment, and general clinical results. Accurate and high-quality imaging is necessary to make accurate diagnoses, efficiently design treatments, and consistently monitor diseases. This review summarizes forty-two research studies on diverse MIQA approaches and their effects on performance in diagnostics, patient results, and efficiency in the process. It contrasts subjective (manual assessment) and objective (rule-driven) evaluation methods, underscores the growing promise of machine intelligence and machine learning (ML) in MIQA automation, and describes the existing MIQA challenges. AI-powered tools are revolutionizing MIQA with automated quality checks, noise reduction, and artifact removal, producing consistent and reliable imaging evaluation. Enhanced image quality is demonstrated in every examination to improve diagnostic precision and support decision making in the clinic. However, challenges still exist, such as variability in quality and variability in human ratings and small datasets hindering standardization. These must be addressed with better-quality data, low-cost labeling, and standardization. Ultimately, this paper reinforces the need for high-quality medical imaging and the potential of MIQA with the power of AI. It is crucial to advance research in this area to advance healthcare.

## 1. Introduction

Medical image quality assessment (MIQA) is a fundamental aspect of medical imaging that significantly impacts diagnostic accuracy, patient management, and overall clinical outcomes. High-quality medical images are crucial for accurate diagnosis, treatment planning, and disease monitoring [1,2,3]. With the continuous advancement of imaging technologies, robust MIQA methodologies are essential to ensure the reliability and efficacy of these technologies in clinical practice. Medical imaging technologies such as X-ray, Computed Tomography (CT), magnetic resonance imaging (MRI), ultrasound, and Optical Coherence Tomography (OCT) have become integral to modern healthcare. These modalities enable the non-invasive visualization of internal body structures, aiding in diagnosing and managing various medical conditions. However, the diagnostic utility of these technologies depends heavily on the quality of the images produced [2,4]. Poor-quality images can lead to misdiagnosis, unnecessary repeat examinations, increased patient anxiety, and higher healthcare costs. Integrating deep learning (DL) and artificial intelligence (AI) technologies in image quality assessment (IQA) represents a significant advancement in the field, offering improved efficiency, consistency, and accuracy in image quality evaluations [5].

Additionally, the integration in IQA involves using DL algorithms within the AI framework to leverage traditional AI methods (like data processing and decision making algorithms) and DL’s ability to analyze high-dimensional data, ultimately improving the accuracy and consistency of image quality evaluations. By distinguishing between these methodologies, researchers may understand that DL is the primary technology driving automation in IQA. At the same time, AI refers to the overall system architecture and approach that may involve additional methods beyond DL. This systematic review synthesizes findings from these studies, providing insights into the current trends and future directions in IQA research. The primary objective of this review is to systematically evaluate the existing literature on MIQA, focusing on the methodologies and technologies used to assess image quality across different imaging modalities. This review aims to summarize the impact of image quality on diagnostic accuracy and patient outcomes, compare subjective and objective assessment techniques, explore the role of AI and machine learning (ML) in MIQA, identify the challenges and limitations in the current MIQA practices, and suggest future directions for research and practice in MIQA. By addressing these objectives, the review seeks to provide a comprehensive overview of the current state of MIQA, highlight effective techniques, and identify areas for further research that can enhance the field. Understanding the strengths and limitations of the existing MIQA methods will help develop better strategies for improving image quality, ultimately leading to better patient care.

The paper is structured as follows: a literature review discussing the importance of medical image quality, different imaging modalities, quality metrics, synthesized findings, and the role of AI and challenges and limitations; a discussion summarizing key points, implications for practice, and future directions; and a conclusion that recaps the main findings, provides recommendations, and concludes with the potential impact of advancing MIQA on healthcare outcomes. The references section lists all the referenced works.

## 2. Methodology

### 2.1. Research Question

The primary research question of this systematic review is to explore the current methodologies and technologies used in MIQA and their impact on diagnostic accuracy, patient outcomes, and operational efficiency. This review will address several sub-questions as follows:What are the subjective and objective techniques employed for MIQA?How do different imaging modalities influence the assessment of image quality?What role do AI and ML play in enhancing MIQA?What are the common challenges and limitations in current MIQA practices?

These questions guide the systematic review, ensuring a comprehensive examination of all aspects of MIQA. By answering these questions, the evaluation seeks to provide an in-depth understanding of the field, highlight the most effective techniques, and identify areas for further research

### 2.2. Study Selection

The selection of studies adhered to rigorous inclusion and exclusion criteria. The inclusion criteria encompassed peer-reviewed articles specifically addressing MIQA; research involving various imaging modalities such as X-ray, CT/PET, MRI, OCT, ultrasound, and echocardiogram; exploration of subjective or objective assessment techniques within MIQA; and articles discussing advancements and applications of AI and ML in MIQA. The exclusion criteria included non-English language articles, studies lacking sufficient data related explicitly to MIQA, and conference abstracts without full-text availability. The study selection process was methodically depicted using a flow diagram in Figure 1, which provided a clear visual representation of the number of studies identified, screened, and ultimately included in the review.

We employed a systematic data harvesting and screening process to minimize bias through our research. Studies with vague or incomplete descriptions were removed, and studies that provided quantitative and reproducible approaches for assessing image quality were preferred. In addition, a sensitivity analysis was conducted to investigate the impact of potential bias on our results, ensuring that our findings were not disproportionately influenced by studies with a high degree of heterogeneity in their quality assessment methods. The distribution of the included studies is portrayed in Figure 2.

By referring to Table 1 and Figure 3, it is noticeable that medical imaging modalities from 2020 to 2024 reveal a clear trend of increasing diversity in imaging techniques, peaking in 2023. To begin with, in 2020 and 2021, only MRI was recorded; however, in 2022, there was a noticeable expansion with the introduction of CT/PET and X-ray imaging, indicating a growing interest in multimodality approaches. As per Figure 3, it suggests a significant rise in IQA-related research and applications. Moreover, MRI remained the most consistently used modality across all the years, reflecting its fundamental role in medical imaging studies.

## 3. Literature Review

### 3.1. Imaging Modalities

#### 3.1.1. X-Ray

X-ray imaging is one of the most used modalities in medical diagnostics due to its availability and cost-effectiveness, making it a staple in clinical settings such as emergency departments, outpatient clinics, and inpatient wards. It is particularly beneficial for diagnosing various conditions, including fractures, infections, and lung diseases. However, the quality of X-ray images can be significantly affected by multiple factors, such as patient movement, improper positioning, and suboptimal exposure settings, which can compromise the clarity and diagnostic utility of the images. Therefore, the importance of MIQA for X-ray images cannot be overstated. MIQA involves a comprehensive evaluation of critical parameters such as contrast, sharpness, and noise levels, essential for ensuring that the photos are of sufficient quality to support accurate diagnoses.

Contrast distortion significantly influences human perception of image quality, often occurring during image acquisition. Practical IQA methods for contrast-distorted images are essential for benchmarking and optimizing contrast-enhancement algorithms. Various techniques, such as histogram equalization, histogram modification, and gamma correction, are employed to enhance the contrast of medical images [20]. These methods help improve the visibility of anatomical structures, which is vital for accurate diagnosis. Objective measures, like the Natural Image Quality Evaluator (NIQE) and Blind/Reference Image Spatial Quality Evaluator (BRISQUE), assess image quality without requiring a perfect reference image. These measures provide quantitative data to compare different processing methods and select the most effective ones, ensuring the enhanced images meet clinical standards.

Using non-reference quality assessment methods like NIQE and BRISQUE is significant as they allow for evaluating image quality in scenarios where a reference image is unavailable. This approach is beneficial in real-world applications where various distortions often degrade images. Studies have demonstrated that these objective measures can effectively support contrast improvement in X-ray images. Researchers and clinicians can use algorithms that do not require reference images to ensure consistent and reliable IQA, facilitating better diagnostic decisions.

Integrating image-text contrastive learning with medical domain knowledge fusion represents a significant advancement in IQA. This method combines the visual features of X-ray images with textual information from diagnostic reports, leveraging large-scale clinical data [19]. The fusion of these data types enhances the predictive accuracy of IQA models, reducing the reliance on radiologists’ subjective evaluations, which the evaluators’ skill level and experience can influence. By aligning the local visual patch features of X-ray images with multiple text features, this approach ensures that the visual features contain more detailed and fine-grained information. This method improves the accuracy of IQA. It extends to other tasks, such as multi-lesion segmentation and disease progression prediction, showcasing its versatility and broad applicability in medical imaging.

#### 3.1.2. Computed Tomography (CT)/Positron Emission Tomography (PET)

IQA is an essential aspect of medical imaging, particularly for PET/CT scans, due to its profound impact on diagnostic accuracy and clinical outcomes. PET/CT imaging is crucial in diagnosing and monitoring various conditions, including cancer, cardiovascular diseases, and neurological disorders. Ensuring high-quality images is vital to providing accurate and reliable diagnostic information, directly influencing patient management and treatment decisions. The selected studies prove that image quality is influenced by resolution, noise, radiation dose, tracer quality, motion artifacts, and scanner calibration. These factors will affect the image clarity, accuracy, and diagnostic effectiveness.

The introduction of DistilIQA [4], a novel distilled vision transformer network, highlights the importance of no-reference IQA for CT images. This approach addresses the challenge of assessing image quality without a pristine reference image, which is particularly useful in clinical settings where reference images are often unavailable. DistilIQA integrates convolutional operations with multi-head self-attention mechanisms, demonstrating superior performance in predicting quality scores for low-dose chest and abdominal CT scans. This method ensures enhanced images meet clinical standards while optimizing radiation doses, thus balancing image quality and patient safety.

Another innovative approach is using self-supervised learning frameworks for no-reference IQA of CT scans. These frameworks leverage self-supervised training strategies to detect virtually inserted objects with geometrically simple forms, enabling the automatic calculation of quantitative quality metrics. This method, known as Deep Detector IQA (D2IQA), has shown robust performance in computing perceptual image quality across different dose levels. The correlation between D2IQA metrics and radiologists’ quality scores further validates its effectiveness, making it a valuable tool for real-time clinical applications [10].

While no-reference IQA methods are crucial, full-reference IQA (FR-IQA) methods also play an essential role. Studies have investigated the applicability of FR-IQA methods, such as peak signal–noise ratio (PSNR) and structural similarity (SSIM), for CT images. Initially designed for natural images, these methods have shown strong correlations with subjective assessments by radiologists, indicating their potential utility in medical imaging. The Visual Information Fidelity (VIF) method has demonstrated the highest correlation with subjective assessments, providing an accurate alternative measure for CT image quality [16].

Applying DL models for automatic IQA has gained significant attention. For instance, a deep learning-driven multi-view multi-task image quality assessment (M^2^IQA) method has been developed for chest CT images. This method utilizes a multi-view fusion strategy and several algorithms tailored for specific evaluation tasks, achieving high precision and sensitivity. Such automated systems reduce the labor-intensive process of manual assessments and mitigate human limitations, such as fatigue and perceptual biases, enhancing the overall efficiency and reliability of IQA [17].

Automated IQA using DL models is also being applied to PET imaging. For instance, an ML classifier has been shown to effectively assess the quality of F18-FDG PET images, providing performance equivalent to manual assessments by experienced radiologists. This approach uses neural networks to classify image quality based on predefined criteria, ensuring consistent and objective evaluations [14]. Similarly, a DL model trained to recognize optimal and poor-quality PET images demonstrated high accuracy, sensitivity, and specificity, further supporting the feasibility of automated IQA in clinical practice [15].

#### 3.1.3. Magnetic Resonance Imaging (MRI)

IQA is a critical component in the field of MRI as it directly influences the accuracy of diagnoses and subsequent treatment plans. The inherent complexity and sensitivity of MRI technology make it susceptible to various artifacts and distortions, such as patient movement and noise, which can degrade image quality and impact the diagnostic value of the images. The need for robust and reliable IQA methods in MRI is paramount to ensure that the photos meet the high standards required for practical medical evaluation and intervention.

One of the significant challenges in MRI is the presence of motion artifacts, particularly in cardiac MR images and fetal brain MRI. For example, a multi-task learning (MTL) based classification model has been proposed to detect different levels of motion artifacts in cardiac MR images. Utilizing k-Space-based Motion Artifact Augmentation (MAA) and a novel compound loss function significantly improves the segmentation accuracy of cardiac structures affected by respiratory motion artifacts [8]. Similarly, a semi-supervised DL method has been developed in fetal brain MRI to detect slices with artifacts during the brain volume scan, improving model accuracy and demonstrating the feasibility of online IQA during scans [6]. These advancements highlight the importance of addressing motion artifacts to enhance the diagnostic reliability of MRI.

Accurate IQA is essential to optimize MRI protocols and ensure that images provide the necessary detail for effective diagnosis. For instance, DL models have been applied to automate image quality assessment in bi-parametric prostate MRI, demonstrating performance comparable to that of less-experienced human readers [7]. Additionally, a Deep Convolutional Neural Network for Image Quality assessment (IQ-DCNN) has been developed to mimic the expert visual assessment of whole-heart MR images, showing strong agreement with human experts and aiding in assessing image quality during reconstruction processes. These examples underscore the role of advanced IQA methods in maintaining high diagnostic standards in MRI.

Integrating DL and AI into IQA for MRI has revolutionized the field by providing automated, objective, and consistent assessments. Various studies have demonstrated the effectiveness of DL models in evaluating MRI quality. For instance, a novel no-reference MRIQA method using deep convolutional neural network architecture has shown superior performance in capturing MR image characteristics and predicting quality compared to state-of-the-art NR-IQA methods [1,13]. Furthermore, an Optimized Deep Knowledge-based No-reference Image Quality Index (ODK-NIQI) has been developed, combining advanced DL techniques to achieve the best performance in denoising MRI images [2]. These advancements illustrate the significant potential of DL in enhancing the accuracy and efficiency of IQA in MRI.

IQA for MRI must also cater to specific clinical requirements, such as evaluating dynamic MRI acquisitions and assessing images generated by advanced techniques like Generative Adversarial Networks (GANs). Tools have been developed to evaluate image quality for conventional and dynamic MRI protocols, providing comprehensive evaluations, including artifact detection and signal–noise ratio (SNR) measurements [9].

The goal of IQA in MRI is to ensure that image quality assessments are consistent and reliable, aligning with the perceptions of experienced radiologists. Extensive studies have been conducted to evaluate the correlation between objective IQA metrics and subjective evaluations by radiologists [12].

#### 3.1.4. Optical Coherence Tomography (OCT)

Optical Coherence Tomography (OCT) is a non-invasive imaging technique extensively used in ophthalmology for detailed visualization of the retina and other eye parts. The accuracy and reliability of OCT in diagnosing eye diseases largely depend on the quality of the captured images. Therefore, robust image quality assessment (IQA) mechanisms are critical to ensure the effective use of OCT in clinical settings.

The assessment of image quality in OCT is pivotal because it directly impacts the diagnostic process. Poor-quality images can lead to incorrect diagnoses or the overlooking of critical conditions. Anterior Segment Optical Coherence Tomography (AS-OCT) is a non-invasive imaging technique that captures high-resolution, cross-sectional images of the eye’s anterior segment, including the cornea, iris, and lens. It is widely used in ophthalmology for diagnosing and monitoring ocular conditions. For instance, the study by Chen et al. [22] discusses the challenges of manual IQA, noting its time-consuming nature and subjectivity, which can result in inconsistent evaluations from the perspective of AS-OCT. Automated IQA systems, leveraging DL techniques, provide a solution to these issues, offering consistent and objective assessment. Chen et al. emphasize the importance of identifying factors that degrade image quality, such as artifacts caused by eyelashes or glare, which are particularly prevalent in images of the anterior segment of the eye. Chen et al. discussed four quality factors that impact OCT images in their study. These factors include ‘Eyelash’, which refers to artifactual hyper and hyperreflective signals within the anterior chamber caused by the presence of eyelashes; ‘Glare’, characterized by a hyperreflective line of the signal resulting from the fixation light used in the AS-OCT camera; and ‘Left-cropped’ and ‘Right-cropped’, which describe instances where the anterior chamber cross-sectional image is cropped at the left or right edge, respectively. The study classifies the overall quality of these images into three levels: ‘Good’, ‘Limited’, and ‘Poor’. A ‘Good’ quality image offers a clear, artifact-free view of the entire anterior chamber. In contrast, a ‘Limited’ quality image has reduced fidelity but still allows for identifying critical clinical features. A ‘Poor’ quality image, on the other hand, is unsuitable for use due to the presence of extensive artifacts.

Furthermore, the research by Chen et al. [21] introduces an unsupervised anomaly-aware framework for Optical Coherence Tomography Angiography (OCTA) image quality assessment. This study highlights the necessity of distinguishing between high-quality and low-quality images, as low-quality images can severely impact the accuracy of diagnostic algorithms. The framework proposed by researchers employed a feature-embedding-based low-quality representation module to quantify and classify image quality, categorizing the images into three image quality levels: outstanding, gradable, and ungradable. This approach is crucial in clinical applications where only high-quality images should be used for diagnostic purposes, while low-quality images require enhancement or are deemed unsuitable.

Moreover, the study on the quality assessment of anterior segment OCT images in [40] underscores the role of automated systems in enhancing clinical workflow. This work identifies specific quality factors such as blurriness, inadequate illumination, and contrast issues that can affect OCT images’ clarity and diagnostic value and are graded on a three-level grading system (good, limited, or poor). By employing automated IQA systems, clinicians can quickly and accurately assess the quality of images, ensuring that only diagnostically viable images are utilized, thereby improving eye care services’ overall efficiency and reliability.

The importance of image quality assessment in OCT images cannot be overstated. High-quality images are essential for accurate diagnosis and effective treatment planning in ophthalmology. As discussed in the referenced works, the development and implementation of automated IQA systems are critical in addressing the limitations of manual assessment methods. These systems enhance the consistency and objectivity of quality evaluations and facilitate identifying and correcting image artifacts, ensuring that the OCT images used in clinical practice are of the highest possible quality.

#### 3.1.5. Ultrasound

Ultrasound is a versatile, real-time imaging modality. Quality assessment parameters include resolution, contrast, and the presence of artifacts such as speckle noise. According to [24], image quality assessment in ultrasound imaging is crucial for ensuring diagnostic accuracy and consistency. The quality of ultrasound images, typically assessed manually by sonographers, heavily influences the reliability of diagnoses. This manual process highly depends on the operator’s skill and experience, introducing variability that can affect diagnostic outcomes.

High-quality images are essential for accurately identifying anatomical structures and pathological conditions, while poor-quality images can obscure critical details due to noise, shadows, and artifacts. Automated systems for image quality assessment, such as the proposed Unsupervised Ultrasound Image Quality Assessment Network (US2QNet), address these challenges by providing objective and reproducible evaluations. These systems enhance the visibility of important features, reduce the burden of manual annotation, and ensure consistent image quality across different operators and institutions. Moreover, they are valuable in training novice sonographers, offering immediate feedback, and helping improve imaging techniques. By streamlining clinical workflows, automated quality assessment systems save time and resources, allowing for more efficient retake of suboptimal images. In research and development, reliable automated quality assessment facilitates the creation of annotated datasets necessary for developing and validating new imaging technologies and algorithms. Overall, the implementation of automated image quality assessment in ultrasound imaging significantly impacts diagnostic accuracy, clinical efficiency, and the advancement of medical imaging technology.

#### 3.1.6. Echocardiogram

As mentioned in [25], assessing image quality in echocardiography is crucial for several reasons. Echocardiography, a widely used non-invasive imaging modality, is essential in diagnosing and monitoring various cardiac conditions. The quality of echocardiographic images directly impacts measurements’ accuracy and diagnoses’ reliability. High-quality images enable the clear visualization of cardiac structures, accurate delineation of myocardial borders, and precise measurement of cardiac function parameters, such as ejection fraction and wall thickness.

Automated image quality assessment also supports research and development in medical imaging. High-quality annotated datasets are essential for developing and validating new imaging technologies and diagnostic algorithms. Computerized systems provide reliable and consistent annotations, critical for training ML models and conducting robust clinical studies. These advancements can lead to the development of more accurate and efficient diagnostic tools, ultimately improving patient outcomes.

Finally, the development and implementation of automated quality assessment systems reflect a broader trend toward using AI in healthcare. These systems leverage advanced algorithms to analyze complex image data and provide insights that enhance clinical decision-making. As AI technology evolves, automated image quality assessment will likely become a standard component of echocardiographic practice, contributing to improved diagnostic accuracy, efficiency, and patient outcomes.

Therefore, the importance of image quality assessment in echocardiography cannot be overstated. Automated systems for assessing image quality offer numerous benefits, including reducing operator dependence, minimizing diagnostic errors, enhancing clinical workflow efficiency, supporting training and education, and facilitating research and development. By ensuring high-quality echocardiographic images, these systems play a critical role in improving the accuracy and reliability of cardiac diagnoses and ultimately enhancing patient care. Figure 4 illustrates the key factors mentioned above.

### 3.2. Quality Metrics

Image quality metrics assess the accuracy, clarity, and fidelity of images. They help evaluate imaging techniques using objective and perceptual measures, ensuring reliability in applications like medical imaging, DL, and computer vision tasks. The selected studies use various image quality assessment techniques, briefly introduced in Figure 5, for better understanding.

#### 3.2.1. Subjective Assessment Techniques

1.Expert Radiologist Review

Subjective assessment often involves expert radiologists reviewing images to evaluate quality based on clinical experience and expertise. Expert radiologist review is a widely used method for assessing image quality. Radiologists evaluate images based on contrast, sharpness, and the visibility of anatomical structures. Their assessments are based on clinical experience and expertise, making them valuable for identifying subtle image quality issues that objective metrics may not capture.

Several studies have investigated the reliability of expert radiologist reviews for MIQA. The following Table 2 shows some of the techniques used by various researchers.

These studies have shown that radiologists can provide consistent and accurate image quality assessments, but there is still some variability between radiologists. A Mean Opinion Score (MOS) is used to overcome these challenges. It averages the scores given by multiple reviewers to achieve a consensus on image quality. This method helps mitigate individual biases and provides a more balanced image quality evaluation. According to the study in [13], MRI images are rated on a subjective scale, and the average scores from multiple reviewers provide an overall quality metric. This method is beneficial when a single expert’s opinion may not be sufficient and a collective judgment is preferred. Nevertheless, MOS has limitations such as subjectivity, observer bias, limited granularity, and difficulty handling complex image quality issues.

2.Visual Grading Analysis (VGA)

VGA is a subjective assessment technique that involves grading images on a predefined scale based on visual quality and diagnostic acceptability. VGA is widely used in clinical practice to evaluate image quality, providing a standardized method for comparing images. The procedures of some selected studies are summarized in Table 3.

Several studies have investigated the reliability and validity of VGA for MIQA, along with the expert radiologist review. These studies have shown that both can provide consistent and reliable image quality assessments, but there is still some variability between observers. Training and calibration can help reduce this variability, but objective metrics are also needed to provide consistent and reliable assessments.

VGA focuses on image quality evaluation through subjective grading based on predefined criteria, assessing attributes such as contrast, noise, and sharpness. It is moderately subjective, with standardized scales helping to improve consistency, and is primarily used in research and image optimization. In contrast, an expert radiologist review involves comprehensive image interpretation to identify pathological findings such as tumors, fractures, and anomalies. This process is highly subjective, relying on clinical expertise and contextual analysis, and is essential for patient diagnosis and treatment planning.

3.Inter Observability and Intra-Observability

Inter-observability concerns the agreement between different observers for assessing medical image quality to ensure other individuals’ consistent use of criteria. Intra-observability refers to the repeatability of an individual observer’s assessments over time [7,29]. High inter- and intra-observability is challenging when human observers are inconsistent or vary in their subjective judgments. Approaches to improving consistency include using standardized protocols, rigorous training, AI/ML programs, and intense learning models like CNNs, which offer the potential to provide objective and reproducible evaluations. These are essential considerations for accurate diagnosis, improved patient outcomes, and reliable quality assessment in medical imaging. To overcome the challenges in this method, the Intraclass Correlation Coefficient (ICC) [27,35,39], the statistical index, was employed. ICC is a statistical measure commonly used to assess the consistency or agreement between multiple observers. It can be applied in binary classification and scoring systems depending on how the image quality is rated.

4.Calibration and Consensus Methods

Calibration and consensus methods are employed to reduce variability among radiologists. These methods involve training sessions and establishing standardized evaluation criteria, ensuring assessment consistency [11]. Calibration sessions help align grading criteria among radiologists for prostate MRI images, minimizing subjective differences and enhancing the reliability of image quality evaluations [16]; the Double Stimulus Continuous Quality Scale (DSCQS) method achieves consensus in CT image assessments, where images are evaluated in pairs to compare quality, providing a more structured approach to subjective evaluations.

These subjective image quality assessment techniques provide valuable insights into the quality of medical images, ensuring they meet the necessary standards for accurate diagnosis. Each method has its strengths and limitations, and the choice of technique often depends on the specific requirements of the study and the type of images being assessed. Combining these subjective techniques with objective measures can enhance the overall image quality assessment and improve diagnostic outcomes. In most instances, the subjective assessments are combined and used for evaluations and annotation of datasets.

#### 3.2.2. Objective Assessment Techniques

Accurate image quality assessment is crucial for ensuring reliable diagnostics and effective treatment planning in medical imaging. Various image quality assessment (IQA) metrics are employed to evaluate the quality of images, particularly in modalities such as chest X-rays, MRI, and CT scans. These metrics are categorized into three main types: full-reference image quality assessment (FR-IQA), Reduced-Reference Image Quality Assessment (RR-IQA), and No-Reference Image Quality Assessment (NR-IQA) [1,2,13,16,20]. However, in some instances, it is categorized as a full-reference image quality assessment (FR-IQA), distribution-based image quality assessment (DB-IQA), and no-reference [12]. 

1.Full Reference Image Quality Assessment (FR-IQA)

FR-IQA methods require a reference image of perfect quality to compare against the assessed image. Full reference methods compare the test image to a high-quality reference image, using metrics such as peak signal–noise ratio (PSNR) and Structural Similarity Index (SSIM).

The following studies have investigated the reliability and validity of full reference methods for MIQA. These studies have shown that full reference methods can provide consistent and reliable assessments of image quality, but they require the availability of a high-quality reference image. 

Signal–Noise Ratio (SNR): This measurement evaluates the clarity of an image by comparing signal strength to background noise [27,28,31,32,33,34].Contrast–Noise Ratio (CNR): CNR focuses on the visibility of structures by measuring the contrast between two regions relative to noise [27,28,31,32,33,34].Structural Similarity Index Measure (SSIM): SSIM measures the similarity between two images by focusing on structural information and assessing luminance, contrast, and structure. It is widely used due to its ability to closely correlate with human visual perception [16,20].The SSIM index is calculated using various image windows—the measure between two windows *x* and *y* of standard size *N* × *N*. Here, *x* is a window from the reference image, and *y* is the corresponding window from the other image.Multi-Scale Structural Similarity Index Measure (MS-SSIM): An extension of SSIM, MS-SSIM evaluates images at multiple scales, providing a more comprehensive analysis by considering luminance, contrast, and structural similarity across different resolutions [16,20].Feature Similarity Index Measure (FSIM): FSIM focuses on the similarity of features between images, particularly those that are perceptually significant. This metric is crucial for assessing the visual quality of images [16,20].Peak Signal–Noise Ratio (PSNR): PSNR measures the ratio between a signal’s maximum possible power and noise’s power that affects the image’s quality. It is a standard metric in FR-IQA, commonly used for evaluating image compression and restoration [16,20].Visual Information Fidelity (VIF): This metric quantifies the amount of information that can be extracted by a human observer from the distorted image relative to the reference image, providing insight into the visual quality loss [16].Information Fidelity Criterion (IFC): Similarly to VIF, IFC measures the fidelity of visual information in the distorted image, offering a quantitative evaluation of image quality degradation [16].Noise Quality Measure (NQM): NQM evaluates image quality by considering noise characteristics, particularly in assessing medical images where noise can obscure essential details [16].Visual Signal–Noise Ratio (VSNR): This metric focuses on the signal–noise ratio in images, which is particularly important in medical imaging to differentiate between meaningful signal and noise [16].Information Content-Weighted SSIM (IWSSIM): A variant of SSIM, IWSSIM weights the structural similarity by the information content, providing a more nuanced assessment of image quality based on content importance [17].

2.No Reference

No reference methods are widely used for the objective assessment of image quality. These methods assess image quality without needing a reference image, using algorithms to evaluate features such as noise, blur, and contrast.

These studies used no reference methods for MIQA. These studies have shown no reference decision score can provide consistent and reliable image quality assessments.

Natural Image Quality Evaluator (NIQE): A reference-free metric that assesses image quality based on natural scene statistics, often used when reference images are unavailable [9].Blind/Referenceless Image Spatial Quality Evaluator (BRISQUE): ML models trained on natural images with known distortions are used to evaluate the quality of a given image. It is effective for various types of distortions [10,12,19,20]Blind Image Quality Assessment (BIQA): Techniques under this category assess image quality without reference images, often leveraging DL approaches to model complex distortions [2].Maximum Mean Discrepancy (MMD): MMD measures the distance between the distributions of authentic and generated images in a feature space, which helps evaluate the consistency of generated images with real ones [9].

3.Reduced Reference

Reduced reference methods use partial information from the reference image to assess quality, striking a balance between full and no reference techniques. These methods use partial information from the reference image to determine quality, balancing full and no reference techniques.

Spatial Efficient-based Entropic Differencing: According to the study by Nikiforaki et al. [30], this metric measures the difference in entropy between reference and distorted images, providing a measure of quality degradation.

4.Distribution-based Metrics

Distribution-based image quality metrics assess the quality of images by comparing statistical distributions of image features between reference (or real) images and generated (or distorted) images. These metrics often focus on the overall distribution of pixel intensities, textures, or other image features rather than individual pixel comparisons. As in [12], the following matrices are distribution-based metrics.

Fréchet Inception Distance (FID): This metric compares the distribution of feature representations of authentic and generated images using the Fréchet distance. It considers the mean and covariance of these features, often extracted using a neural network trained on a large dataset like ImageNet [9].Kernel Inception Distance (KID): Like FID, KID uses the Inception network to extract features from images and compares these using polynomial kernel Maximum Mean Discrepancy (MMD). It has the advantage of providing unbiased estimates even with small sample sizes.Inception Score (IS): IS uses the output distribution of a classifier (usually InceptionV3) to evaluate the diversity and quality of generated images by comparing the distribution of predicted labels with a uniform distribution.These metrics are widely used to assess the quality of images generated by GANs. They enable the comparison of statistical properties in a deep feature space, ensuring that generated images closely resemble real ones. Operating in feature space rather than raw pixels makes them perceptually relevant.

#### 3.2.3. DL Model-Based Approaches 

Medical imaging is manually assessed by technicians or radiologists, posing subjective and time-consuming quality evaluation. The DL models, which replace manual inspection or support it in these approaches, can improve the efficiency of image quality evaluation and ensure the evaluation outcome’s stability [13]. IQA is essential for accurate diagnostics and effective treatment results [1]. DL methods are considered valuable for determining what factors may affect image quality by providing automated and objective image assessments. As predicted, the integration of AI in health will optimize the workflow of clinical practice, improve patient outcomes, and change the current delivery system of healthcare. However, the data quality gap is a significant bottleneck from model development to its clinical application in medical AI research. Many solutions relied on already overburdened medical practitioners to perform data classification, often resulting in slow progress [23].

The following models are frequently used in these selected studies:

As mentioned in the methodology of [21], they have proposed an automated IQA framework based on multi-task learning. Convolutional neural networks (CNNs), such as VGG, ResNet, and DenseNet, are feature extractors that learn shared knowledge from images. Therefore, VGG16, ResNet18, ResNet34, ResNet50, and DenseNet121, VGG16 achieve the best performance.

The study by Stępień et al. [1] introduced a method for enhancing MRI quality assessment (MRIQA) by fusing multiple DL architectures. The approach leveraged their feature extraction and classification strengths by integrating well-known models such as VGG, ResNet, and Inception. Utilizing transfer learning with networks pre-trained on ImageNet, the final classification layers are replaced with regression layers to tailor the models for quality prediction. The process involved three steps. Namely, feature extraction to MRI images is processed by each network to extract relevant features, Feature Fusion to the features from the various networks are combined and input into a Support Vector Regression (SVR) module, and quality prediction to the SVR model maps these combined features to quality scores. The method adopted network layers to MRI characteristics and employed a radial basis function kernel in SVR for regression. The results demonstrated that this fusion approach, including networks like DenseNet-201, GoogLeNet, and ResNet, significantly improves quality prediction. Additionally, DeepDream visualizations reveal that the fused networks better manage distortions and provide more detailed feature responses than single networks.

The proposed technique, Multi lEvel and multi-model Deep Quality Evaluator of MR Images (MEDQEMRIs) in [13], fused two DL networks of different complexities that belong to the same family. Re-input data are fed to ResNet18, Resnet,50, and their fusion. The main finding was that the quality assessment is performed by a high-level quality model trained on scores of quality models obtained for layers of the networks.

In [23], which is known as DeepFundus, InceptionResNetV2 architecture was used for model construction, trained with TensorFlow, Keras, and early stopping to prevent overfitting. This addressed the data quality gap and achieved Areas Under Curve (AUCs) over 0.9 in image classification concerning overall quality, clinical quality factors, and structural quality analysis on both the internal test and national validation datasets. The value achieved signifies that the model accurately distinguishes between image quality categories. Specifically, there is a higher chance that the model will correctly identify the image quality category.

The DL model in [11] is InceptionResnetV2 on the development set using probabilistic prostate masks and an ordinal loss function. The research examined how well a DL model checked the ability for bi-parametric prostate MRI. It compared the model’s results to what experts agreed on and to less experienced readers. This revealed to us that the DL model does about as well as less-experienced readers in judging quality. The study concluded that DL models, trained on more typical datasets with input from more experts, could give reliable automatic quality checks. They might even help with or replace people looking at images to judge quality.

In assessing CT image quality, a study [17] introduced M^2^IQA, which consisted of a fusion of YOLOv8 and U-Net. This retrospective study analyzed chest CT images from 327 patients, using 1613 images from 286 patients for model training and validation, while 41 patients were reserved for ablation, comparative, and observer studies. The M^2^IQA method, driven by DL, utilizes a multi-view fusion strategy across three scanning planes (coronal, axial, and sagittal) to evaluate image quality for tasks like inspiration, position, radiation protection, and artifact assessment. It achieved 87% precision, 93% sensitivity, 69% specificity, and a 0.90 F1-score on an additional test set. Comparably, [37] also mentioned the DL-based method, which combines image selection, tracheal carina segmentation, and bronchial beam detection. The score obtained by this method is compared with the MOS given in the observer study.

The study [15] developed a DL model using DenseNet to assess PET image quality, encompassing acquisition, preprocessing, and training with data augmentation and cross-validation. The DL-based assessment tool reliably categorized PET images into “Good” or “Poor” quality and provides detailed image and scanning information, supporting clinical research through accurate quality assessment.

The study introduced Deep Detector IQA (D2IQA) [10], a new CT image quality assessment system that copies radiologists’ work by spotting fake lesion-like objects affected by different noise levels. D2IQA used a self-taught Cascade R-CNN model trained with made-up lesions of various shapes, sizes, and contrasts, eliminating the need for accurate labels. The system showed strong results, showing object size, contrast, and noise impact detection accuracy, and it kept a strong link with what radiologists thought across different dose levels. Compared to old-school full-reference (FR-IQA) and no-reference IQA (NR-IQA) measures, D2IQA does better in picking up on quality changes. It worked well across different body areas and artifact types. This new approach looks set to push forward CT image quality checking and improvement, with plans to expand its use to other kinds of medical imaging in future studies.

In [6], the study introduced an enhanced semi-supervised learning approach for fetal brain MRI quality assessment, integrating region of interest (ROI) and consistency into the Mean Teacher Model. The model employed DL architecture based on ResNet-34 for student and teacher networks. The student network is trained with a combined loss function consisting of classification loss and consistency loss between the student and teacher networks. The teacher network’s parameters were updated using an exponential moving average. This method introduced ROI consistency loss to target the brain ROI, which ensured that the network focused on brain features by comparing features from masked and original images. The approach also used conditional entropy for further refinement. It showed more significant benefits with smaller labeled datasets and confirmed the importance of the additional regularization terms. This novel approach enhanced MRI quality assessment and offers potential for integration with fetal motion tracking algorithms to optimize imaging workflows. Similarly, the work mentioned in [36] the development of MD-IQA, which consisted of multi-scale distribution regression to reduce prediction uncertainty and improve the robustness in reducing prediction uncertainty. The study used a vision transformer (ViT) and CNN modules to extract global and local features to enhance the representation capability.

According to the Schwyzer et al. [14], two experienced readers scored image quality from 1 (not applicable for diagnosis) to 4 (best for diagnosis). Their scores helped train a ResNet-34 DL model built with the fast.ai library on 400 images split into training, validation, and test groups. The main takeaways included the classifier’s trustworthy assessment of image quality, performance matching standard SNR measures, and steady results across different reconstruction settings.

The work in [8] study presented a multi-task DL model that analyzes cardiac MRI. It focuses on classifying image quality and segmenting cardiac structures. The classification part sorts images into three motion artifact levels. It also had a side job of guessing the breath-hold type during imaging. The model tweaked a 3D ResNet-18 structure and uses a multi-task loss. For segmentation, they picked a 3D U-Net setup with a mixed loss. This proves they work well in tackling motion artifacts in heart MRIs.

According to the study in [9], automated methods using image quality metrics and statistical analyses were explored while addressing human assessment limitations. Experts’ ratings and reaction times indicated sensitivity to image quality, with FID, MMD, and NIQE showing good correspondence, particularly for lower-quality images. A deep-quality assessment model best captures subtle differences in high-quality photos. The study recommended combining group analyses, spatial correlations, and distortion and perceptual metrics for comprehensive evaluation.

The GAN-guided nuance perceptual module [42] (G2NPAN) system used a GAN to assess the quality of medical fused images by integrating several advanced techniques. The Generator, a convolutional neural network with down-sampling and up-sampling phases, converted a fused image into a high-quality version using residual blocks and LeakyReLU activations. The Discriminator, another CNN, distinguished between authentic and generated images using LeakyReLU and mean absolute error. The Unique Feature Warehouse (UFW) extracted and integrated spatial features from the images at multiple scales, focused on subtle details critical for medical assessments. Additionally, the Attention-Based Quality Assessment Network (AQA), built on VGG11, evaluates the quality of the fused image by comparing it to the high-quality reference image produced by the GAN. This network used an attention mechanism (Class Activation Mapping—CAM) to enhance interpretability and focus on significant features.

Q-Net [25] architecture was a multi-stream, multi-output regression model designed for multi-labeled predictions, specifically targeting anatomical features in medical images. It used a combination of spatial and temporal modules, with each spatial module consisting of several convolutional layers and an LSTM layer for temporal feature extraction. The model employed Rectified Linear Unit (ReLU) activation, batch normalization, and dropout to prevent overfitting. It was trained using a 5-fold cross-validation with data augmentation, aimed for real-time application with optimal performance, memory efficiency, and inference speed. The architecture was compared with other state-of-the-art models like DenseNet121, ResNet, and VggNet, focusing on quality attributes using mean absolute error as the cost function.

At the training stage, unsupervised anomaly-aware framework (UNO-QA) [21], they trained an encoder with multi-scale pyramid pooling and multiple decoders for multiple scales with only outstanding samples. All the testing samples were fed into a low-quality representation module at the inference stage. After that, excellent and non-outstanding samples were classified. For the non-outstanding samples, they extracted and concatenated the output features of the decoders and then applied feature dimension reduction and clustered to subdivide no-outstanding samples into gradable and unreadable samples. To assess the adaptability of this framework, they analyzed different anomaly detection models, PaDim, PatchCore, and Fastflow, which were incorporated into this pipeline. They have observed that their low-quality representation module combined with hierarchy clustering achieved the best classification performance.

The deep convolutional neural network for automated image quality assessment (IQ-DCNN) [7] inputs 2D patches from 3D image volumes in the axial, sagittal, and coronal orientations. The network comprised four convolutional layers, three fully connected layers, and a final regression layer to output a quantitative image quality score. It employed an anti-bias L1 loss function to align the predicted grades with ground truth, compensating for non-uniform grade distribution. Inspired by the VGG-16 model, the architecture was optimized using three-fold cross-validation on a dataset of 424 scans, divided into training/validation (324) and test sets (100). Dropout regularization prevented overfitting, and patch quality grades were averaged per patient for a final assessment. The IQ-DCNN demonstrated performance comparable to human intra- and interobserver agreement, achieving an R^2^ of 0.78 and a kappa coefficient of 0.67, indicating substantial agreement with human experts in assessing image quality. The model effectively tracked image quality during compressed sensing reconstruction, aligning closely with expert evaluations. The IQ-DCNN successfully mimicked expert assessments of 3D whole-heart MR images and could automatically compare different reconstructed volumes. However, increasing dataset size and diversity, implementing data augmentation, or fine-tuning hyperparameters can be incorporated to achieve higher consistency with expert evaluations.

Optimized Deep Knowledge-based NIQI (ODK-NIQI) [2] for MRI image quality assessment used a three-step approach involving deep images before creating a diverse denoised image database, noise removal, and feature extraction from noisy and denoised images. It employed a ConvNet model enhanced by shuffle shepherd optimization and mish activation, with weighted average pooling to consolidate results. Based on a pre-trained VGG-16 model, the improved deep knowledge algorithm refined hyperparameters and enhanced feature extraction. The ODK-NIQI method outperforms traditional NIQI techniques, demonstrating superior performance and consistency across standard metrics such as SROCC, RMSE, MAE, and PLCC.

In an UnSupervised learning-based Ultrasound image Quality assessment Network (US2QNet) [24], a Variational Autoencoder (VAE) was trained using reconstruction loss to learn feature representations from pre-processed ultrasound images, optimized for both the reconstruction of images and the latent space distribution. The VAE was enhanced with a clustering module to improve the quality of feature representations. It jointly optimized the VAE’s reconstruction and clustering loss to better align image features with quality clusters. The validation of urinary bladder ultrasound images demonstrated that the proposed framework could generate clusters with 78% accuracy and perform better than state-of-the-art methods.

As in DistilIQA described in [4], the model evaluated CT image quality using a distillation-based vision transformer. It combined a vision transformer network, which merges self-attention with convolutional operations, and a distillation setup. A “teacher networks” group passed on knowledge to one “student network” in this setup. The primary goals were training and testing the model on different CT datasets, including CT scans of the chest and abdomen, and predicting image quality scores. The model’s structure has a convolutional stem and a vision transformer.

In the work of [5], the Swin transformer is described. In other words, the Shifted Window Transformer is a vision transformer known for its hierarchical and efficient image processing capabilities. Unlike traditional CNNs, Swin transformers excelled at capturing long-range dependencies within an image. They achieved this through a hierarchical structure that processes images at multiple scales, allowing the model to capture local details and broader contextual information. The “shifted window” mechanism involved dividing the image into non-overlapping windows and then shifting these windows in subsequent stages. This method ensured that the model captured information from neighboring windows, offering a more comprehensive image understanding. Similarly, the authors of [38] proposed a quality control method based on YOLOV8 and Convolutional Block Attention Module (CBAM) and Swin transformers. As specified by the authors, the suggested model can be utilized in evaluating CT phantom images and holds the potential for setting a new standard in quantitative assessment methodologies.

The Swin transformer-based Multiple Color-space Fusion network (SwinMCSFNet) classifiers [41] detected subtle abnormalities and differentiated between various tissue types, which is crucial for accurate medical diagnoses. SwinMCSFNet was particularly useful in medical applications. Its design allowed it to handle the complexity and high dimensionality typical of medical images, often with fewer parameters than traditional CNN-based methods, yet achieved superior or comparable performance. They had focused on implementing a Multiple Color-Space Fusion network to integrate representations from various color spaces.

Semantic Aware Contrast Learning (SCL) [3] is an advanced technique in the field of ML, beneficial for tasks that require distinguishing between subtle differences in data, such as IQA in medical imaging. The core idea behind SCL is to enhance the ability of a model to differentiate between classes by focusing on semantic features, which are meaningful and relevant attributes that capture the essence of the data. In the context of IQA, SCL involves training a model to recognize the general quality of an image and the specific semantic content that may influence the quality assessment. This approach was particularly beneficial in medical imaging, where specific anatomical structures or pathological features can significantly impact the perceived quality of an image.

The work in [29] mentioned the design of a two-stage dual-task network framework, which takes the fetal MRI slices as input and outputs the image quality label. This framework included two stages: brain localization and dual-task with brain segmentation and quality assessment tasks. The model consisted of a brain localization module using U-Net for coarse segmentation, followed by a dual-task module with a feature extraction network, segmentation head, and quality assessment head, using complex parameter sharing for efficient fetal brain MRI analysis. The process typically involved creating a contrastive loss function that encouraged the model to push apart representations of images with different quality levels while pulling together representations with similar quality levels. By incorporating semantic awareness, the model can learn to prioritize features most relevant for assessing image quality, such as clarity, presence of artifacts, and contrast between different tissues. Table 4 summarizes the IQA methods used in their studies, and Table 5 on DL techniques in medical imaging.

## 4. Results

### 4.1. Characteristics of Included Studies

Many MIQA studies employ subjective and objective evaluation methods, often leveraging DL models. Subjective assessments play a significant role in evaluating these models, with automated modules predominantly relying on DL techniques. Key subjective evaluation methods include expert radiologist review and visual grading assessments, which provide critical insights into the effectiveness of MIQA systems.

In MIQA research, two primary approaches are utilized for objective assessment: full reference and no reference methods. Full reference methods compare the quality of processed images against a pristine reference image, offering a comprehensive evaluation but requiring access to high-quality reference images. On the other hand, no reference methods assess image quality without direct comparison to a reference. This poses challenges in accurately gauging image fidelity but is more practical in scenarios where pristine images are unavailable or impractical.

However, no reference assessments generally yield lower accuracy than full reference methods due to the inherent difficulty in establishing a baseline standard for image quality. This limitation underscores the ongoing challenge of developing robust evaluation techniques to reliably assess MIQA without requiring extensive reference data.

A recurring issue highlighted across many studies is the scarcity of resources, which encompasses both the availability of high-quality reference images and the computational resources necessary for implementing sophisticated DL models. These resource constraints present significant hurdles in advancing MIQA research, as they limit the scale and complexity of experiments that can be conducted, thereby influencing the development and validation of DL-based MIQA systems.

### 4.2. Impact of Image Quality

Medical imaging modalities like X-ray, CT/PET, MRI, OCT, ultrasound, and echocardiogram are vital for diagnosis and treatment planning. Each modality requires specific image quality assessment (IQA) techniques to ensure diagnostic accuracy. X-ray imaging benefits from methods like histogram equalization and gamma correction to enhance contrast, while non-reference IQA methods such as NIQE and BRISQUE provide objective quality assessments. In CT/PET, DistilIQA and Deep Detector IQA enable accurate, reference-free assessments. MRI faces challenges like motion artifacts; advanced DL methods, including IQ-DCNN and the Visual Information Fidelity (VIF) metric, help maintain high diagnostic standards. OCT imaging relies on automated systems to address artifacts, enhancing the clarity of retinal images. Ultrasound and echocardiography, used extensively in cardiology and obstetrics, benefit from automated IQA systems like US2QNet, which provide consistent, operator-independent assessments. These IQA techniques are crucial for optimizing image quality, reducing diagnostic errors, and improving patient care across various clinical settings.

### 4.3. Use of Artificial Intelligence and Machine Learning

AI and ML have begun a new chapter in MIQA, bringing in computer-driven methods that are causing a revolution in how we check and improve medical images. These tools play a key role in automating jobs like judging image quality, cutting down noise, and fixing flaws, which are crucial to making correct medical diagnoses and planning treatments.

Studies in this area show that AI and ML algorithms can make MIQA more accurate and productive. It ensures that high-quality medical images are essential for reliable interpretation and analysis. For instance, research comparing how AI algorithms and radiologists evaluate CT image quality reveals that AI gives steady and trustworthy assessments [17]. These algorithms reach accuracy levels matching those of skilled human experts, which hints at their ability to standardize quality checks across medical imaging practices. As per the [26], a DL-based AI algorithm was developed to classify the image quality of prostate MRI, showing that higher-quality T2-weighted images (T2WIs) were linked to more accurate predictions of extracapsular extension (EPE) in final pathology.

DL models, including CNNs such as VGG, ResNet, and DenseNet, have been popular in developing IQA systems [8,11,15]. Meanwhile, implementing a multi-task learning framework with CNNs and the fusion of multiple architectures like VGG, ResNet, and Inception has significantly enhanced the assessment of MRI quality by expanding their strength in feature extraction and classification [13]. Implementations like Deep Fundus [11,23] have addressed specific challenges, such as data quality gaps in image classification. Additionally, models like M^2^IQA [17] and the work in [37], D2IQA [10] focus on assessing CT image quality using advanced architectures like YOLOv8, U-Net, and Cascade R-CNN. Unsupervised approaches, including the Unsupervised anomaly-aware framework [19] and US2IQA [24], employ anomaly detection and clustering techniques for quality categorization. Furthermore, GANs have been used to generate high-quality images and distinguish between real and synthetic data, with models like the GAN-guided nuance perceptual module (G2NPAN) [42] demonstrating effectiveness in this area. The advent of vision transformers (ViTs) and Swin transformers [5,23] marks a significant advancement, offering enhanced capabilities in handling long-range dependencies and complex feature extraction. These innovations in AI and ML are optimizing clinical workflows and improving patient outcomes, ensuring that medical image assessments are more reliable and efficient. The performance evaluation of different approaches of MRI, other imaging modalities (X-ray, OCT, ultrasound, and echocardiogram), and CT are stated in Table 6, Table 7, and Table 8, respectively.

### 4.4. Challenges and Limitations

MIQA using DL holds significant promises for enhancing the evaluation and interpretation of medical images. However, the field faces numerous challenges and limitations that impede developing and implementing robust, generalizable methods. By clustering these limitations into distinct categories, we can better understand and address them, paving the way for advancements in MIQA research.

1.Dataset Quality and Metadata

One of the primary challenges in MIQA research is the quality and comprehensiveness of the available datasets. The absence of metadata in public datasets, such as those used in the PI-CAI challenge, prevents researchers from verifying whether scans adhere to established standards like PI-RADS. This limitation hinders the ability to ensure that the images used for training and evaluation meet the necessary quality and diagnostic criteria, thus affecting the reliability of the resulting DL models [11].

Furthermore, the current MIQA tools often support a limited range of image formats, primarily focusing on DICOM. This restriction limits their applicability to other widely used formats, such as NifTI or ITK MetaImages, which are crucial for broader adoption in diverse clinical settings. The lack of support for multiple formats restricts the usability of these tools and prevents leveraging a wide range of imaging data [30].

Additionally, many studies rely on artificially simulated distorted images that may not accurately reflect clinical photos. This limitation, highlighted in the [16], underscores the need for realistic datasets that closely mimic clinical conditions to improve the applicability and validity of MIQA methods. In addition, environmental noise, patient conditions, and technical limitations of imaging devices affect image quality [42].

2.Standardization and Assessment Scales

The lack of standardized quality assessment scales is another significant limitation in MIQA research. For instance, the prostate MRI feasibility study used a 3-point Likert scale for image quality assessment. While applicable for specific study designs and model training, such arbitrary scales do not provide a standardized approach to image quality assessment. The absence of universally accepted and standardized scales limits the comparability of results across different studies and hampers the development of universally applicable MIQA methods [11,19].

Moreover, the non-uniform distribution of quality grades in datasets poses a challenge. As observed in the study on whole-heart MR images, the uneven distribution of quality grades necessitates anti-biasing methods to avoid prediction bias. Balanced and well-distributed datasets are crucial for training and evaluating accurate MIQA models [7].

3.Observer Variability and Consensus

Observer variability is a well-recognized challenge in MIQA. Many studies rely on grading a single reader from a single center, limiting the findings’ generalizability. For example, the survey of whole-heart MR images was based on grading from a single reader at a single center. This approach does not account for variability in grading practices across different readers and centers. Multi-reader, multi-center studies are essential for developing more generalizable and reliable MIQA models [7].

Intra- and interobserver variability further complicates the situation [29]. Significant variability among observers, particularly in grading intermediate-quality datasets, introduces uncertainty. This variability was noted in both human readers and DL models, reflecting increased uncertainty and emphasizing the need for methods to reduce observer variability in MIQA.

4.Model Design and Generalization

Designing optimal deep convolutional neural network (DCNN) architecture involves navigating numerous hyperparameters and configurations, making it complex. The study on whole-heart MR images pointed out the difficulty of performing exhaustive searches for the best-performing models, given the vast number of potential configurations. Advanced optimization techniques and automated network design methods are needed to address this complexity and enhance model performance [7]. The generalization of unseen data remains a critical concern in MIQA research. Models trained on specific datasets may not perform well on different types of medical images. This limitation, highlighted in various studies [13,36], emphasizes the importance of developing robust models capable of generalizing across diverse imaging modalities and clinical scenarios. Furthermore, Bos et al. [31] and Drews et al. [34] discussed that the small sample size, the lack of transparency in the commercial algorithm, and the need for comparison with other DL approaches will also affect the limitations in generalization.

5.Evaluation Metrics and Cost Constraints

Challenges in objectively determining the impact of poor-quality images on diagnostic outcomes limit the ability to validate and refine DL models. This issue was evident in the prostate MRI study, where the influence of image quality on diagnostic performance was not evaluated due to the difficulty of objective assessment [11]. High costs associated with labeling datasets present another significant limitation. The “DistilIQA” study identified these costs as a barrier to using more accurate ground truth metrics. This cost constraint highlights the need for cost-effective and efficient labeling methods to advance MIQA research [4]. As mentioned in the study by Treder et al. [9], a noise-to-image model could enhance metrics quality based on reference images without exploring an image-to-image model.

6.Dataset Size and Diversity

As mentioned in [14], small dataset sizes and limited metrics reduce the robustness and generalizability of MIQA research. The study on the optimized deep knowledge-based no-reference image quality index for denoised MRI images highlighted these limitations, suggesting that larger, more diverse datasets and improved metrics are necessary for advancing the field [2]. However, the study in [9] reveals that a controlled lab environment potentially reduces internal validity, with varying display devices used by participants possibly confounding group analyses

By clustering these limitations, we can identify key areas that require attention and improvement in MIQA research. Addressing these clustered limitations through targeted strategies can enhance the development and application of DL methods in medical image quality assessment. This approach will involve improving dataset quality and metadata, standardizing assessment scales, reducing observer variability, optimizing model design, ensuring model generalization, developing cost-effective labeling methods, and expanding dataset size and diversity.

Table 9 shows the limitations of the studies presented in [2,4,7,9,11,13,14,16,19,29,30,31,34,36].

## 5. Discussion

AI and ML have drastically improved the domain of MIQA, with DL frameworks, particularly convolutional neural networks (CNNs), driving the standardization and automation of image quality evaluations. These innovative approaches have dramatically improved the capacity to evaluate image quality for various imaging modalities, such as X-rays, CT/PET scans, MRI, and ultrasound, as illustrated in Figure 6.

The use of DL has been consistently referenced across multiple years, with a significant increase in studies in 2023, as per the tabular data in Table 10. GANs started gaining attention in 2022 and saw continued development in 2024. Classification techniques have steadily increased, especially in 2023, aligning with the broader trend of applying supervised learning for MIQA. Semi-supervised learning was explored in 2020 and 2024, suggesting that researchers are investigating alternative learning paradigms for MIQA. Vision transformers (ViTs) and Swin transformers appeared notably in 2022 and 2024, indicating a shift towards transformer-based architectures for medical imaging tasks. Furthermore, the categorization in the following table is based on learning paradigms and functional objectives. Therefore, GANs are distinct in their approach, even though they fall under unsupervised techniques.

However, many obstacles still exist to exploit MIQA’s potential completely. One of the most significant problems is the scarcity and heterogeneity of datasets, which restricts the possibility of evaluating image quality in various clinical situations. Further, the absence of image quality assessment metrics standardization only increases the difficulty in comparing results from varying studies and systems. Several reasons cause standard quality evaluation scales to be absent in MIQA research. Since each imaging modality has unique artifacts and quality issues, it is challenging to establish a standard scale across several of them.

Furthermore, the subjective character of image quality evaluation, which primarily depends on radiologists’ assessments, introduces variances in experience and viewpoint. Many studies have historically used arbitrary scales like the Likert scale, which hinders the transition to a standardized approach. Dataset limitations, including inconsistent quality annotations, further exacerbate this issue. Nevertheless, as AI-driven MIQA models evolve, new quality assessment metrics emerge, making standardizing even more challenging.

Furthermore, human observers’ subjective nature of image quality assessment can lead to inconsistencies and variability of results. A further significant limitation comes from artificially generated data and the prohibitively expensive nature of data annotation, which are obstacles to the broader deployment of such methods in practical scenarios. Despite such challenges, advancements in techniques like GAN and ViTs continue to improve the robustness and accuracy of MIQA systems. This development has wide-ranging potential to enhance the diagnostic accuracy of medicine and the overall realm of diagnostic medicine. It will be necessary to address data quality challenges, standardize evaluation metrics, and reduce data labeling costs to guarantee the continuous evolution of MIQA and its integration into clinical practice. The combination of AI and ML has significantly advanced the field of MIQA, specifically the application of DL models, such as CNNs, for the automation and standardization of the image quality assessment processes. Incorporating these novel methods has improved assessing image quality for various imaging modalities, including X-rays, CT/PET, MRI, and ultrasound. However, the field of MIQA faces several challenges. These include restricted availability and diversity of datasets to gauge the quality of an image, a lack of standardization in quality assessment scales, and difficulty from observers in evaluating image quality.

Methods for the clinical relevance of automated IQA systems can be used at full in clinical workflows to improve workflow efficiency, robustness, and quality, accomplished mainly by systems integration. Automated IQA solutions can support radiologists in assessing the quality of an image in real-time, indirectly save time from manual assessments, and ensure that only high-quality photos are used for diagnosis. Automated assessments notwithstanding, manual reviews must be used with experienced radiologists because they are better prepared to deal with complex or borderline cases.

There should be a balance between automated scoring and manual reviews. If radiologists have automated systems that flag low-quality images for subsequent review, they can better triage these cases appropriately. Combining past evaluation techniques and DL approaches can achieve a synergistic performance. Meanwhile, expert evaluations or grading scales provide the context and fine-grained judgment that DL models would likely fail to account for. The pairing of both structures creates a more solid and precise system, fulfilling a higher quality assurance in medical imaging alongside proper diagnosis suggestions.

Additionally, reliance on synthetically generated data and associated data annotation costs constitute essential bottlenecks. Despite these challenges, advances in techniques such as GANs and ViTs continue to support the robustness and accuracy of systems for medical image quality assessment. This research progress holds promise for improving the accuracy of medical diagnostics and diagnostic medicine. Improving data quality, standardization, and lowering data labeling costs are key challenges for future medical image quality assessment progress.

Nevertheless, some doubts need to be addressed, such as how MIQA models are more generalized across different imaging modalities and clinical environments. Can multi-modal learning improve the accuracy and robustness of MIQA? Which role can self-supervised or few-shot learning play in reducing the reliance on large, annotated datasets? The integration of MIQA models seamlessly into real-world clinical workflows.

## 6. Conclusions

IQA is crucial in improving diagnostic decisions by ensuring that medical images are clear, accurate, and suitable for clinical interpretation. High-quality images provide enhanced visibility of anatomical structures, abnormalities, and lesions, leading to more precise detection and diagnosis. By identifying and addressing issues such as noise, resolution problems, or artifacts, IQA methods reduce the likelihood of misinterpretation and diagnostic errors. Automated IQA systems ensure consistent image quality, which helps standardize results across different clinicians and healthcare settings. Moreover, reliable IQA helps optimize clinical workflows by minimizing the need for repeat imaging, saving time and resources. Effective IQA methods ultimately contribute to more accurate, consistent, and efficient diagnostic decision-making, improving patient outcomes.

Based on the integration of AI and ML technologies, mainly CNNs, and augmented with DL models, the MIQA field has undergone a significant transformation. These new methodologies aim to make medical image quality assessment more objective, automatable, and reliable than human quality assessment across diverse medical imaging modalities while replacing the time-consuming MIQA methods. However, many challenges exist to be overcome, such as the need for high-quality, expansive datasets and standardized quality assessment scales and addressing observer variability. The reliance on simulated datasets and outsourcing labeling tasks to mitigate labeling costs are barriers to generalizing the methodology to areas beyond simulation. Overcoming these obstacles will demand strategic actions to enhance the dataset’s quality and availability, develop consistent evaluation procedures, and improve DL models’ generalization performance. Addressing these barriers is crucial for creating the MIQA field, promoting reliable and consistent assessment of medical images, and ultimately enhancing diagnostic outcomes and patient care. 

## Figures and Tables

**Figure 1 jimaging-11-00100-f001:**
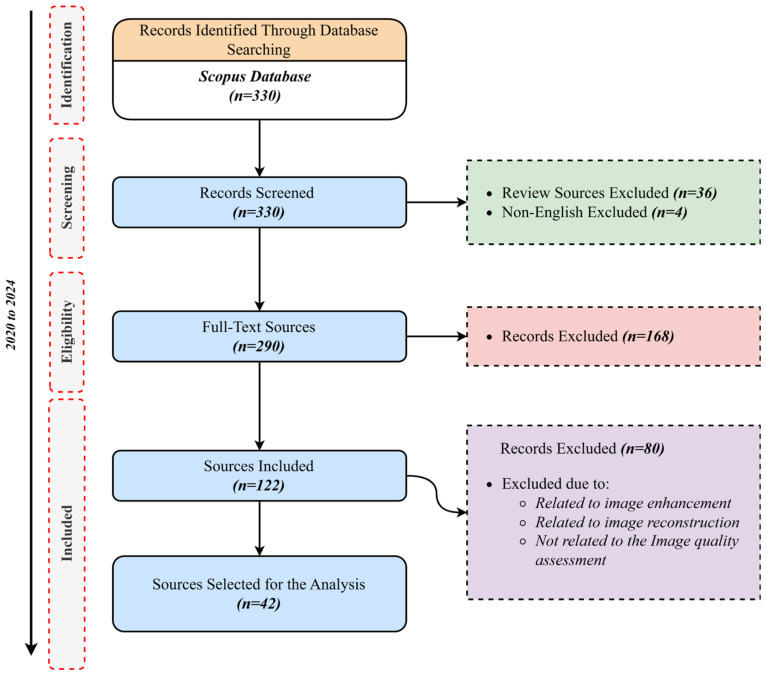
The PRISMA model for the study selection.

**Figure 2 jimaging-11-00100-f002:**
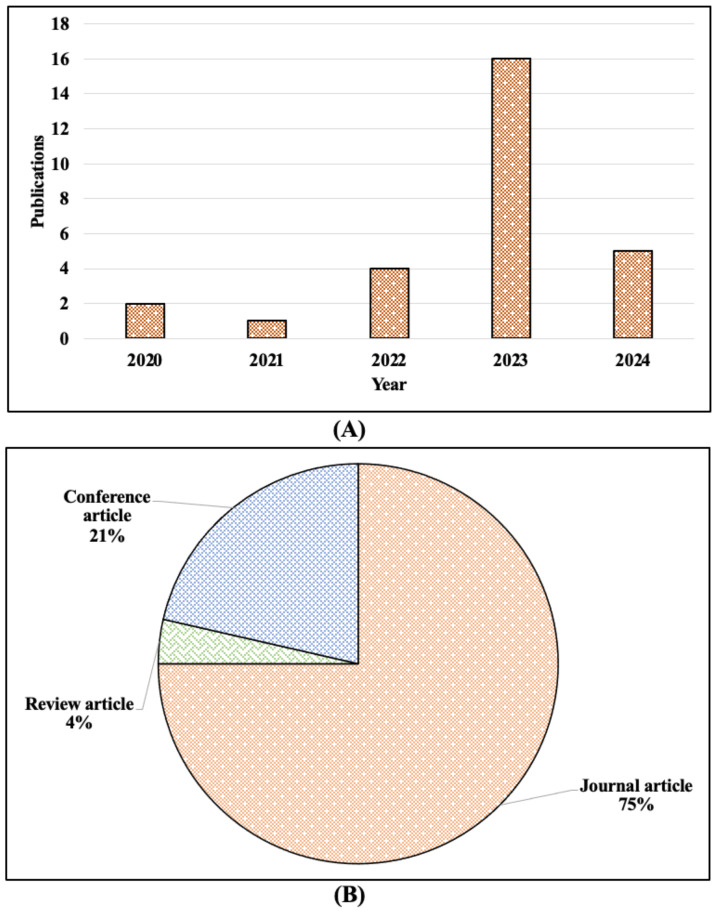
Source selection distribution: (**A**) year-wise publication in medical image quality assessments; (**B**) source type distribution in the selected studies.

**Figure 3 jimaging-11-00100-f003:**
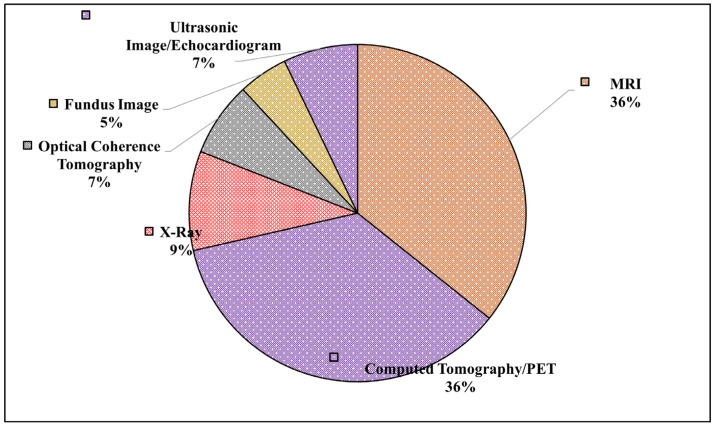
Medical imaging technologies that used IQA in previous studies.

**Figure 4 jimaging-11-00100-f004:**
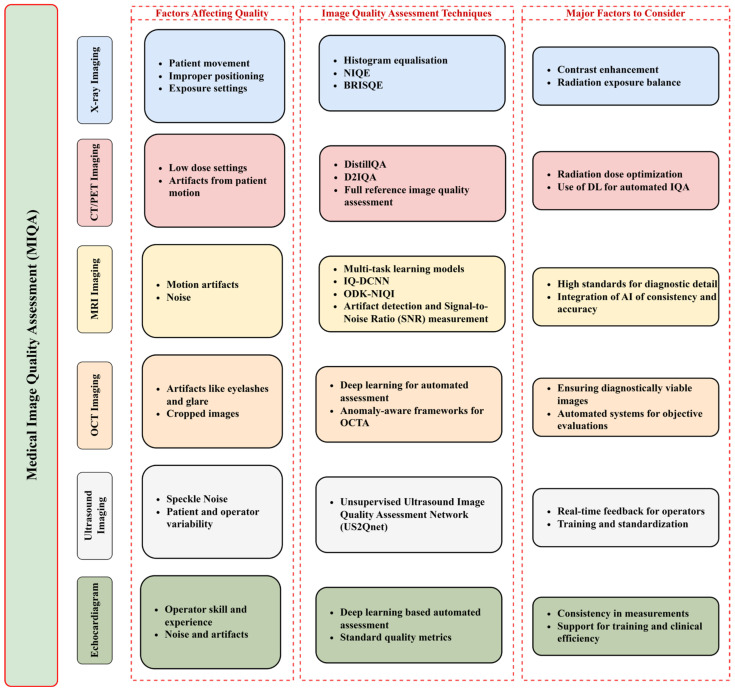
Imaging modalities and IQA techniques.

**Figure 5 jimaging-11-00100-f005:**
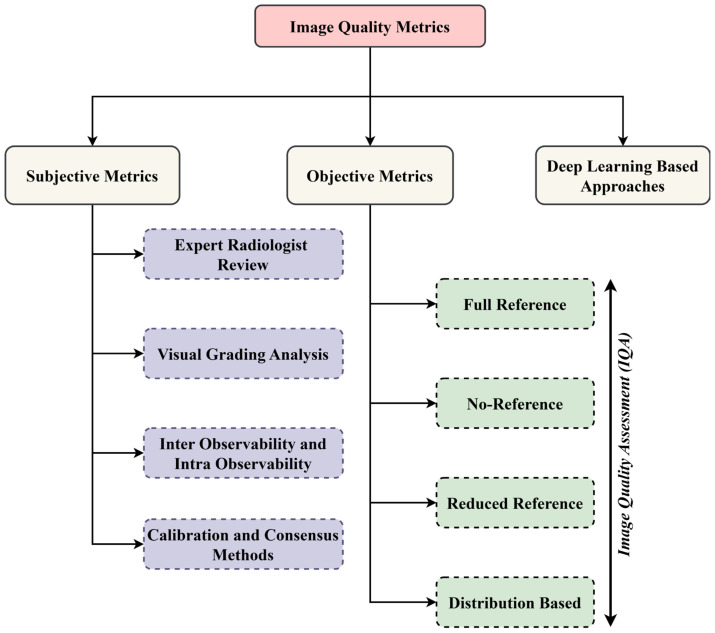
Categorization of image quality metrics.

**Figure 6 jimaging-11-00100-f006:**
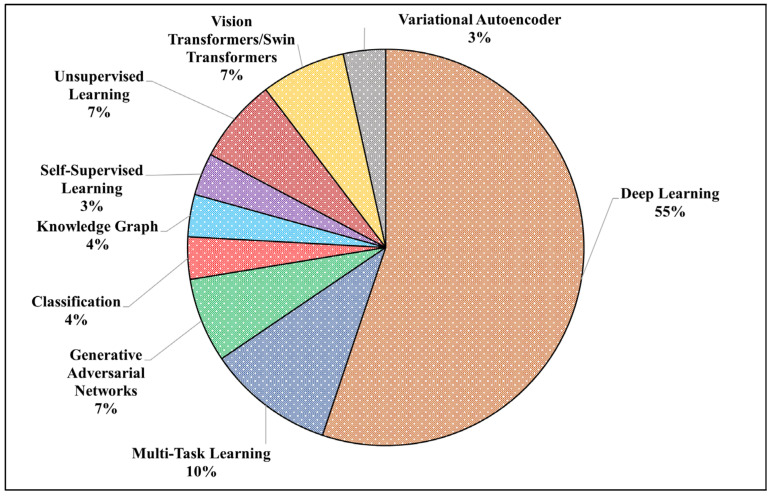
AI methodologies used in previous research for image quality assessments.

**Table 1 jimaging-11-00100-t001:** Imaging methodologies used by previous studies as year-wise distribution.

Year	MRI	CT/PET	X-Ray	OCT	Fundus Image	Ultrasonic Image/Echocardiogram	Other
2020	[6,7]						
2021	[1]						
2022	[5,8,9]	[10]	[5]				
2023	[2,11,12,13]	[14,15,16,17]	[18,19,20]	[21,22]	[23]	[3,24,25]	[3]
2024	[26,27,28,29,30]	[4,31,32,33,34,35,36,37,38,39]		[40]	[41]		[42]

**Table 2 jimaging-11-00100-t002:** Strategies implemented in studies involving expert radiologist review.

Author(s)	Technique
[3]	Utilized two experts to grade retinal images, focusing on aspects like blurring and uneven illumination, which can affect the diagnosis of retinal diseases.
[7]	Whole-heart MR images were reviewed for artifacts and vessel sharpness, essential for evaluating coronary artery disease.
[8]	Radiologists evaluated cardiac MR images for motion artifacts, which can significantly impair the diagnostic quality, particularly in detecting myocardial conditions.
[10]	Three radiologists used a 10-point scale to score CT images, offering a detailed image quality assessment from multiple perspectives.
[11]	It involved fetal brain MRI images, where radiologists categorized images based on their diagnostic utility, which is vital in prenatal assessments.
[14]	PET images were classified into five quality grades, providing a nuanced view of image usability for diagnostic purposes.
[15]	PET/CT images were assessed using a scoring system that evaluates diagnostic value, which is crucial for accurate tumor localization and staging.
[18]	Expert radiologists used a detailed scoring system for chest CT images, focusing on factors such as image noise and sharpness, which are critical for diagnosing pulmonary conditions.
[21]	Discusses grading anterior segment OCT images. The review focuses on the visibility of key anatomical structures like the anterior chamber, which is critical for diagnosing conditions like glaucoma.
[23]	Fundus images were classified based on position, illumination, and clarity, which are essential for detecting ocular diseases.
[25]	Echocardiogram images were evaluated based on the clarity of anatomical features, which is essential for assessing cardiac function and pathology.
[35]	Most images were of average to good quality, with a balanced distribution between low and high-quality classes in the head, neck, chest, and pelvis regions. Still, lower quality was more prevalent in the chest-abdomen interval and abdomen regions.

**Table 3 jimaging-11-00100-t003:** Strategies implemented in studies involving VGA.

Author(s)	Technique
[4]	CT images were assessed using VGA, with categories from “Bad” to “Excellent”, focusing on factors like contrast and resolution, which are crucial for detecting subtle pathologies.
[6]	Fetal brain MRI images were classified as diagnostic, non-diagnostic, or lacking a brain region of interest, aiding in prenatal diagnosis.
[15]	PET images were scored from 1 to 5, considering factors like noise and lesion visibility, which are critical for accurate diagnosis and treatment planning.
[17]	Chest CT images were categorized as “acceptable” or “inacceptable” based on their clarity and the ability to visualize critical anatomical details, which is crucial for diagnosing lung diseases.
[24]	Used a 5-level quality assessment scale for ultrasound images, grading them on parameters like clarity and noise, affecting abnormalities’ detection.
[27]	The 4-point Likert scale was used to measure image quality, diagnostic confidence, noise levels, artifacts, and sharpness of images.
[28]	Overall image quality was rated in the 5-point scoring system.
[31]	A 5-point Likert scale was used only on overall image quality, and specific aspects of image quality, such as artifacts, were not considered.
[32]	Image sharpness and diagnostic confidence were rated using a 5-point Likert scale.
[33]	A scale of scores ranging from -3 to +3 was used to grade the visibility.
[34]	Images were categorized by assigning 1 to the better image and 2 to the worse, with equal quality indicated by a score of 1 for both. Then, each image was evaluated using a 5-point Likert scale: “excellent”, “completely acceptable”, “mostly acceptable”, “suboptimal”, and “unacceptable”.
[39]	CT images were assessed on a scale of none—score 0; minimal—score 1; mild—score 2; moderate—score 3; and severe.
[40]	Categorized OCT images into ‘Good’, ‘Limited’, and ‘Poor’ based on the visibility of retinal layers, which is crucial for diagnosing retinal diseases.
[41]	Retinal fundus images were graded into “Good”, “Usable”, and “Reject” based on the visibility of retinal structures like the optic disk and blood vessels, necessary for conditions like diabetic retinopathy.

**Table 4 jimaging-11-00100-t004:** Summary of the literature review (keyword categorization for image quality assessment in medical imaging).

Author(s)	IQA/Quality Control	Medical Imaging	No-Reference Assessment	Full-Reference Assessment	Perceptual Image Quality
[1]	X	X			
[2]	X	X			
[3]	X	X			
[4]	X	X	X		
[5]	X	X			
[6]	X	X			
[7]		X			
[8]		X			
[9]		X			
[10]	X	X	X		X
[11]	X	X			
[12]	X	X			
[13]	X	X			
[14]		X			
[15]	X	X			
[16]	X	X		X	
[17]	X	X			
[18]		X			
[19]	X	X			
[20]	X	X	X		
[21]		X			
[22]	X	X			
[23]	X	X			
[24]		X			
[25]	X	X			
[26]	X	X			
[27]		X			
[28]		X			
[29]	X	X			
[30]	X	X	X	X	
[31]		X			
[32]	X	X			
[33]		X			
[34]	X	X			
[35]	X	X			
[36]	X	X			
[37]	X	X			
[38]		X			
[39]		X			
[40]	X	X			
[41]	X	X			
[42]	X	X			

**Table 5 jimaging-11-00100-t005:** Summary of the literature review (keyword categorization for DL techniques in medical imaging).

Author(s)	DL	Multi-Task Learning	GAN	Classification	Knowledge Graph	Self-Supervised Learning	Unsupervised Learning	ViT/Swin Transformers	VAE
[1]	X								
[2]	X								
[3]									
[4]								X	
[5]								X	
[6]									
[7]									
[8]	X	X							
[9]	X		X						
[10]						X			
[11]	X								
[12]									
[13]									
[14]									
[15]	X			X					
[16]									
[17]	X	X							
[18]	X								
[19]									
[20]					X				
[21]							X		
[22]	X	X							
[23]									
[24]							X		X
[25]	X								
[26]									
[27]	X								
[28]									
[29]									
[30]									
[31]									
[32]									
[33]									
[34]	X								
[35]	X								
[36]									
[37]	X								
[38]	X								
[39]	X								
[40]									
[41]								X	
[42]			X						

**Table 6 jimaging-11-00100-t006:** Performance evaluation of DL models in MRI.

MIQA Method (MRI)	SRCC	KRCC	PLCC	Root Mean Squared Error	Accuracy	AUC	Kappa Scores	Weighted K Score
Fusion of networks [1]	0.74	0.56	0.81	0.47				
ODK-NIQI [2]	0.98		0.99	1.05				
3D ResNet 18 [8]					0.7			
Semi-supervised learning method [6]					0.85	0.9		
DL based method [11]							0.42	
IQ-DCNN [7]			0.72					0.67
Fusion of DL architecture [13]	0.88	0.72	0.91					
Two-stage dual-task deep learning framework [29]					0.86	0.81		

**Table 7 jimaging-11-00100-t007:** Performance evaluation of DL models in X-ray, OCT, ultrasound, and echocardiogram.

DL Approach	Accuracy	AUC	Kappa Scores	Weighted K Score	Precision	Sensitivity	Specificity	F1-Score
Fusion model [20]					0.96			0.95
DL approach [22]					0.872			0.88
Unsupervised approach [21]	0.72		0.53					
US2Qnet [24]	0.78							
Q-net [25]	0.96							

**Table 8 jimaging-11-00100-t008:** Performance evaluation of DL models in CT/PET imaging.

MIQA Method (CT)	SRCC	KRCC	PLCC	Root Mean Squared Error	Accuracy	AUC	Kappa Scores	Weighted K Score	Precision	Sensitivity	Specificity	F1-Score
DistilIQA [4]	0.87	0.97	0.97									
D2IQA [10]	0.9		0.9									
M^2^IQA [17]									0.87	0.93	0.69	0.9
DL based [14]						0.98				0.89	0.94	
DL based [15]					0.77					0.83	0.71	
MD-IQA [36]	0.97	0.91	0.97									
DL based [38]					0.92							
DL based [37]					0.92		0.75			0.85	0.91	

**Table 9 jimaging-11-00100-t009:** Limitations identified in selected studies.

Study	Limitations
[2]	Acknowledged limitations related to dataset size and metrics used, this study suggested future research to address these areas for improved robustness and applicability.
[4]	Using the Structural Similarity Index (SSIM) as a proxy for radiologists’ scores may not fully capture perceptual nuances critical for clinical assessments. Additionally, the high costs of labeling datasets restricted the use of more precise ground truth metrics, and focusing solely on the number of convolutional blocks without varying their internal architecture may have led to suboptimal performance.
[7]	The study relied on high-quality reference images graded by experts, necessitating anti-biasing methods due to non-uniform quality distribution. Challenges included designing optimal DCNN architectures and hyperparameters, focusing on anatomical structure quality and coronary vessel sharpness limiting generalizability, and training based on grading from a single reader at a single center, introducing intra- and interobserver variability.
[9]	Experimented online instead of in a controlled lab environment potentially reduced internal validity, with varying display devices used by participants possibly confounding group analyses. However, within-subject analyses helped mitigate this issue, and another limitation was the study’s focus on a noise-to-image model, which did not explore an image-to-image model that could enhance metrics quality based on reference images.
[11]	The study faced limitations, including an inability to evaluate the impact of image quality on DL models’ diagnostic performance for clinically significant prostate cancer (csPCa) due to the retrospective nature of the PI-CAI challenge dataset and the lack of metadata for verifying PI-RADS adherence. Additionally, using a 3-point Likert scale for image quality assessment and excluding dynamic contrast-enhanced (DCE) images restricted the scope and novelty of the quality assessment.
[13]	The study highlighted the potential for performance improvement by integrating MRI-specific knowledge into model architecture. While adapting object recognition network structures through joint training enhanced performance, it emphasized incorporating MRI-specific information for further advancements.
[14]	The retrospective nature and small cohort size of this study limited its generalizability. The subjective nature of image quality assessment by readers also posed a challenge, necessitating further validation and optimization for broader clinical applicability.
[16]	The study suggested that full-reference image quality assessment (IQA) methods often require parameters optimized for natural images, which may not be suitable for medical images. The uneven distribution of subjective assessment scores for distorted images could disproportionately affect results, and artificially simulated distortions may not accurately reflect clinical images, limiting the generalizability of the findings.
[19]	The study focused on digital medical X-ray images, utilizing images from the Kaggle database, and acknowledged that environmental noise, patient conditions, and technical limitations of imaging devices could result in poor image quality. It also highlighted the lack of standards for low-contrast x-ray images and emphasized the need for consistently applying multiple methods to enhance contrast.
[29]	Limitations in the study are annotation bias, dataset size, boundary issues, instability, and potential for improved image quality assessment.
[30]	The application supported only DICOM image formats, limiting its usability, with plans to extend support to formats like NifTI or ITK MetaImages, albeit with incomplete DICOM header tag support. It is not designed to evaluate 4D datasets, such as diffusion-weighted imaging, which are important for diagnosis but pose unique image quality assessment challenges.
[31]	The study included manual noise measurement, lack of phantom comparison, and rater bias.
[34]	The study’s limitations included a small sample size, a lack of algorithm transparency, and no comparative analysis.
[36]	The work proposed a method with a dual-branch alignment network to enhance feature extraction capabilities. This will predict quality scores by a multi-scale regression approach.

**Table 10 jimaging-11-00100-t010:** AI methodologies used by previous studies for image quality assessments.

Year	DL	Classification	Multi-Task Learning	Unsupervised Learning	Self-Supervised Learning	Semi-Supervised Learning	Knowledge Graph	VAE	GAN	ViTs/Swin Transformers
2020	[7]					[6]				
2021	[1]									
2022			[8]		[10]				[9]	[5]
2023	[2,11,12,13,14,17,23,25]	[15]	[17,22]	[21,24]			[20]	[24]		
2024	[36,37,38]					[36]			[42]	[4,41]

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
