# Peer review of "A Systematic Review of Medical Image Quality Assessment"

_2313-433X, 2025, doi:10.3390/jimaging11040100_

Round 1
Reviewer 1 Report
Comments and Suggestions for Authors
This manuscript deals with image quality assessment techniques on medical imaging diagnostic systems, especially on CT, MRI, PET-CT, US and OCT. Echocardiography is also included in this study. The abstract describes in short, the systematic review methodology the authors followed in order to complete their research. The Introduction section describes the significance of image quality assessment tests and its impact on diagnosis’s accuracy. The Methods section describes in a simple and understandable manner the review and literature sampling methodology the authors used as well as the statistical analysis and processing they adapted. In the Literature section the authors describe thoroughly any image quality assessment technique that was used during 2020-2024 for CT, MRI, PET-CT, US,OCT and Echocardiography systems according to their literature sample. Perhaps a short paragraph explaining the reasons that hinder any acquisition of images of high quality from the aforementioned imaging modalities (metal-artifacts, patient movement, respiration, hardware integration, image reconstruction process, etc) could enhanced manuscript’s scientific value. Quality metrics section emphasized on image quality assessment based on AI techniques due to the fact that these methods need no reference imaging data. In Results section Table 5 is difficult to be red. Perhaps the authors could make the table bigger, or find another table representation. Sections Discussion and Conclusions focuses on AI based image quality assessment techniques, their benefits and limitations. Bibliography could be characterized poor; however systematic reviews of image quality assessment techniques are rare in the literature.
Author Response
We would like to thank the reviewer for their valuable feedback and insightful suggestions, which have certainly helped improve the quality of our manuscript. Below are our detailed responses to the points raised:- Addition of a Paragraph on Factors Hindering High-Quality Imaging :Here we have mentioned the challenges in hindering high-quality images under challenges and limitations(Section 4.4) in Dataset Quality and Metadata.
- Clarification of the Table 5 Layout: The table has been splitted as Table 6 and Table 7 in order to be legible and increase clarity.
- Bibliography: We appreciate the comment on the bibliography and understand that systematic reviews on image quality assessment techniques are relatively rare in the literature. We have revised the Bibliographysection by adding more relevant and recent references from 2020-2024 (14 more references) to enrich the literature review. While there is limited work on systematic reviews specifically in this area, we have made efforts to include the most pertinent studies to provide a thorough background for our research.
Reviewer 2 Report
Comments and Suggestions for Authors
This paper provides a systematic review of Medical Image Quality Assessment (MIQA) research, encompassing traditional subjective methods and objective metrics such as SSIM, PSNR, NIQE, and BRISQUE. Particular emphasis is placed on automated evaluation methods leveraging artificial intelligence (AI) and deep learning, including CNN, GAN, and Vision Transformer models, specifically designed for detecting and assessing medical imaging modalities such as X-ray, CT/PET, MRI, OCT, ultrasound, and echocardiography. Through tables and figures, the authors extensively summarize and compare the applicable scenarios, evaluation metrics, and performance of various methods, providing a comprehensive literature review and directions for future research.
However, several issues require attention and revision:
(1) For a review article, the current number of references cited is insufficient. Furthermore, the criteria for literature inclusion, exclusion reasons, and processes for assessing literature quality are described superficially, potentially affecting the rigor of the review. It is recommended to provide a more detailed explanation of the literature selection process, complemented by additional flowcharts or tables as necessary.
(2) Although the paper contains several tables and figures comparing different methods, some figure captions lack sufficient detail, limiting the reader’s understanding of key information. It is recommended to enhance the textual explanations accompanying the figures and tables.
(3) In summarizing various methodologies, the paper predominantly lists existing achievements and technical metrics, with limited discussion on their practical clinical limitations, applicability, and underlying causes, thereby lacking critical analysis. It is advisable for the authors to expand the discussion regarding the limitations of existing methods.
(4) Could the authors expand the discussion section to address the clinical applicability of evaluation methods? It would be beneficial to clarify how automated quality assessment systems integrate into clinical workflows and address the balance between automated assessments and manual reviews. Additionally, it would be valuable to discuss the possibility of integrating traditional evaluation methods with deep learning techniques.
(5) In the discussion on the application of deep learning to image quality assessment, can the authors further discuss dataset size, diversity of samples, annotation standards, and data quality considerations?
Reviewer 3 Report
Comments and Suggestions for Authors
This manuscript provides a comprehensive review of Medical Image Quality Assessment (MIQA), exploring various assessment methodologies, their impact on diagnostic accuracy, and the integration of AI and machine learning techniques. Below, I have outlined my major and minor revision suggestions to enhance the clarity, comprehensiveness, and readability of your work.
Major revisions
Line 45: The manuscript states that deep learning is combined with AI technology. However, AI is a broad field that encompasses deep learning. It would be helpful to clarify how these two technologies are being integrated or to explain their distinctions for better understanding.
Lines 86-87: The exclusion criteria are numerous. It is necessary to provide explanation for selecting these specific criteria.
Lines 108-109: The sentence "As per Figure 3, it suggests a significant rise in imaging research and applications" is difficult to understand. Please clarify how Figure 3 demonstrates a significant rise in imaging research and applications.
Lines 144-146: The relationship between reliable image quality assessment and better diagnostic decisions should be explanation further. Explain how image quality assessment methods contribute to improved diagnostic decision.
Line 158 (Section 3.1.2): A brief explanation of CT/PET imaging should be included, along with factors that influence CT/PET image quality.
Line 269: Please provide an explanation of AS-OCT to help readers unfamiliar with this imaging technique.
Lines 375-380: The advantages of MOS (Mean Opinion Score) are well described. However, a discussion on its limitations would provide a more balanced perspective.
Lines 382-391: Both VGA and Expert Radiologist Review involve experts assessing image quality. Please explanation on the differences between these two methods.
Line 395: The manuscript mentions that image quality is assessed based on agreement among different observers. Clarify whether this agreement is based on a binary classification (e.g., "good" or "bad") or a scoring system.
Line 557: Provide a detailed explanation of AUC, including what an AUC value above 0.9 signifies in the context of the study.
Lines 568-570: The results of M2IQA are presented, but additional context is needed to enhance understanding.
Line 663: The R² value (0.78) and kappa coefficient (0.67) are given. Include a discussion on how these values could be improved.
Lines 825-837: Provide reasons why standardized quality assessment scales are lacking.
Table 7: GAN and Unsupervised Learning fall under the broad category of deep learning. Clarify why they are listed separately and provide an explanation of the criteria used to categorize methods in this table.
Minor revisions
Line 45: IQA is used as an abbreviation, but the full term should be provided upon first mention.
Figure 1: The meaning of "sources: 28" at the bottom right corner should be clarified.
Table 1: The title text at the top is rotated, making it difficult to read. Please adjust it for better visibility.
Lines 442-443: The meaning of x and y should be clarified.
Lines 505-515: Provide an explanation of Inception Distance to aid readers unfamiliar with this metric.
Table 4: The title at the top is not easily readable. Adjust the formatting to improve visibility.
Line 901: The manuscript states that Unsupervised Learning was explored in 2022 and 2024, but Table 7 does not reflect this. Ensure consistency between the text and table.
Round 2
Reviewer 2 Report
Comments and Suggestions for Authors
The authors have revised the paper according to the reviewers' comments, and I have no further comments.
Reviewer 3 Report
Comments and Suggestions for Authors
Based on the revisions and the overall quality of the work, I believe the paper is now suitable for publication.
Comments on the Quality of English LanguageBased on the revisions and the overall quality of the work, I believe the paper is now suitable for publication.